# The formation of Haumea and its family via binary merging

Benjamin Proudfoot 🅳 [1✉] & Darin Ragozzine 🅳 [1]

Dozens of families of asteroids in the asteroid belt have similar orbits and compositions because they formed through a collision. However, the icy debris beyond the orbit of Neptune, called the Kuiper Belt, contains only one known family, the Haumea family. So far, no self-consistent explanation for the formation of the Haumea family can match all geophysical and orbital characteristics of the family without invoking extremely improbable events. Here, we show that the family is adequately explained as the product of a merging binary near the end of Neptune's orbital migration. The unique orbital signature of a merging binary, which was not found in extensive searches, is effectively erased during the final stages of migration, providing an explanation for all aspects of the Haumea family. By placing the formation of the Haumea family in the broader context of solar system formation, we demonstrate a proof-of-concept model for the formation of Haumea.

[1] Department of Physics and Astronomy, Brigham Young University, N283 ESC, Provo, UT 84602, USA. ✉email: benp175@gmail.com

 1

Studies of the asteroid belt reveal that asteroid families are most commonly the result of catastrophic collisions between two bodies. In catastrophic collisions, the target is gravitationally disrupted, ejecting collisional family members outwards at velocities large compared to the escape velocity of the asteroid, but small compared to their heliocentric orbital velocity. Despite collisional families being common among the asteroids, only one family is known in the Kuiper belt. The Haumea family, first discovered in 2007[1], was originally hypothesized to be the product of a catastrophic collision, much like the known asteroid families. However, a catastrophic collision like those that form asteroid families cannot be supported by the observations of Haumea for three reasons. First and foremost, the distribution (in semi-major axis, inclination, and eccentricity [$a$-$e$-$i$] space) of family member orbits is ~20 times too small[2,3]. Typical catastrophic collisions impart a change in velocity, $\Delta v$, of a few times the escape velocity of the largest remnant, in the case of Haumea, a $\Delta v$ of ~2000 m s$^{-1}$ [3,4]. The family's current velocity distribution is ~100 m s$^{-1}$ [3,5]. Indeed, for collisions large enough to have debris detectable by current surveys of the Kuiper belt, the spread in proper orbital elements should be comparable to the whole Kuiper belt[6]. Second, the size distribution of Haumea family members is very shallow, with most of the mass concentrated in the largest objects, inconsistent with any kind of catastrophic disruption event[3,7]. Third, these size distributions allow us to estimate that the original mass of the family is a few percent of the mass of Haumea[7], in contrast to tens of percent ejected in a typical catastrophic asteroid collision[4]. Various hypotheses for the formation of the Haumea family have been proposed to explain the small spread in orbital elements[8–11], but all hypotheses that rely on a catastrophic collision are inconsistent with the data[7]. Indeed, even the destruction of an object with the total mass of the known family members (not including Haumea) would already produce a spread in orbits well beyond what is currently seen.

One of the most promising non-catastrophic formation mechanisms is the graze-and-merge collision proposed by Leinhardt et al.[9]. In this mechanism, two large (~650 km) objects suffer a grazing collision at low velocity creating a rapidly spinning body which then sheds mass due to excess angular momentum, forming both the Haumea family and Haumea's two satellites. This mechanism readily creates a compact low-mass family, made primarily from the water ice mantle of already differentiated impactors. The creation of nearly pure water ice family members, consistent with their spectra[12,13] and albedos[14,15], is a strong geophysical constraint on family-formation hypotheses that is well-matched by graze-and-merge style impacts. In this scenario, as family members are ejected due to an excess of angular momentum, their ejection vectors will lie along a tight plane. The previous work[3] showed that this type of ejection does indeed have a detectably unique correlation in $a$-$e$-$i$ space, but ruled out a planar ejection at the ~2.5-$\sigma$ level. Additionally, a slow graze-and-merge collision between two independent large bodies in the excited part of the Kuiper Belt suffers from an extremely low probability[16].

The low probability of an independent collision can be circumvented, if Haumea was originally a binary, probably near-equal mass with each body with a radius of ~650 km, where the components eventually collide in a graze-and-merge style collision, as originally suggested by Marcus et al.[6] Kozai cycles, a dynamical effect that allows a binary to exchange angular momentum between the binary eccentricity and inclination, combined with tidal friction[17,18] (Kozai cycles with tidal friction (KCTF)) is a natural mechanism for explaining this collision although other mechanisms are possible[19], such as encounters with Neptune, geophysical evolution[20], and others.

While the probability of having a near-equal (mass ratio >~10%) binary with a total mass near that of Haumea is not well studied, we view it as plausible, based on formation models and comparisons to other large objects. Formation models[21] show that large ~equal-mass binaries are capable of forming from the gravitational collapse of pebble clouds, though the proto-Haumea binary would occupy the upper mass range of these models. Other models show that a near-equal mass binary could survive implantation into the dynamically excited population of the Kuiper belt[22]. Comparison with other objects provides another indication that large binaries are plausible. Triton, currently a moon of Neptune, is hypothesized to have been a large near-equal binary from the same parent population as the Kuiper belt[23]. The Pluto–Charon system is ~4 times larger than the proposed proto-Haumea binary, with a mass ratio that is amenable to a graze-and-merge type collision formation. It has recently been proposed that the Pluto–Charon system was formed in a similar manner, where the destabilization of a binary system allows for a far higher collision probability[19]. Despite the indications that a near-equal binary proto-Haumea is plausible, the occurrence of large, near-equal mass binaries should be explored further.

With the probability of the graze-and-merge collision thus addressed, we turn to the question of why the observed Haumea family does not exhibit the expected $a$-$e$-$i$ correlation from planar ejecta. The previous work[3] showed that this planar ejecta distribution would survive dynamical interactions for 4.5 GYr and is inconsistent with the observed family at the 2.5-$\sigma$ level. However, these dynamical interactions assumed the planets were in their current orbits and did not place the Haumea family in the context of solar system formation which includes a long (~100 MYr) final stage of Neptune migration. Previous studies speculated that any Neptune migration would likely destroy the tight clustering of family members[3,11]. This led to the supposition that Haumea must have formed after Neptune's migration was completely over, even though age estimates can only say that the Haumea family is >~1 GYr old.

In this work, we show that this assumption is not supported by using migration simulations that show that the compact nature of the Haumea family can be maintained during the late stages of Neptune migration proposed by other investigations. Our simulations additionally reveal that during Neptune migration, the orbital distribution of family members is mixed so that an originally planar family can appear very similar to the family seen today. With this fact in mind, we propose that the proto-Haumea formed as a near-equal binary in the primordial trans-Neptunian belt. Following the standard formation model for the dynamically excited Kuiper Belt[24], the proto-Haumea was first scattered onto a dynamically unstable orbit, captured into one of Neptune's mean motion resonances (MMRs), and subsequently dropped out of resonance near its current orbit. The strong processes in this dynamical excitation and depletion event are too chaotic to expect that the Haumea family formed in the primordial trans-Neptunian belt and was then placed into its observed tight cluster. While we do not specifically propose the MMR from which Haumea was dropped out of, there are several low-order MMRs that could have placed Haumea in its current position (e.g., 3:1, 5:2, 9:4, etc.). The large change in Haumea's heliocentric inclination during this process could have naturally initiated Kozai cycles. KCTF leads to a merger of the proto-Haumea binary; this can take thousands to millions of years depending on the conditions[18,25]. The graze-and-merge collision puts too much angular momentum into the proto-Haumea, which sheds a small amount of mass in the form of icy bodies from its tips. This explains Haumea's near-critical rotation rate, two near-coplanar moons, the small mass of the family, its shallow size distribution, and the low ejection velocities required to form a compact family,

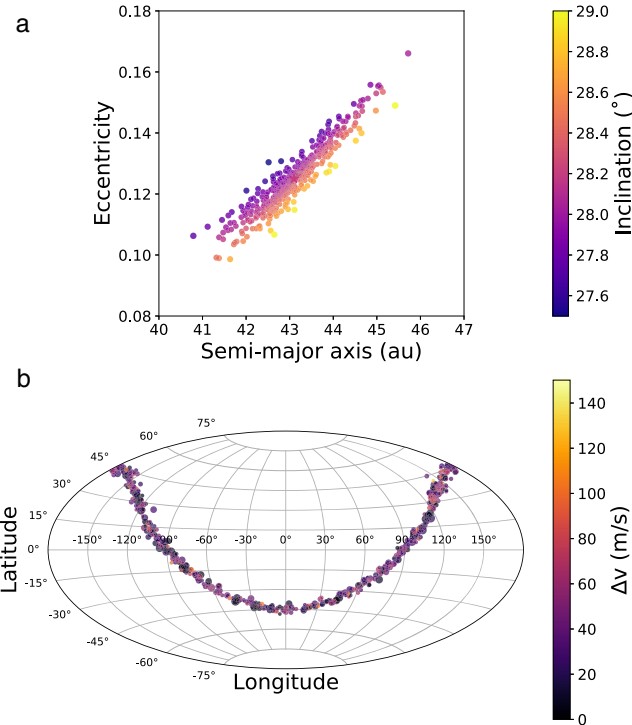

**Fig. 1 A synthetic planar family.** A realization of a graze-and-merge (planar) Haumea family. In panel **a**, the family is shown in *a-e-i* space. The family shape and distribution are typical for a graze-and-merge collision, with the planar distribution of family members visible as a distinct correlation between semi-major axis, eccentricity, and inclination. In panel **b**, the ejection direction of each family member is shown in ecliptic latitude and longitude. Here the size of each family member corresponds to the size of the point. This demonstrates that the family members are ejected in a planar manner, with a typical planar dispersion of ~2°, consistent with the properties of a graze-and-merge family. Source data for this figure are provided as a Source data file.

as have been shown in other studies[3,7,9]. The final part of Neptune migration, especially a jump of eccentricity like that already proposed[26,27], then mixes the objects into the presently observed orbital configuration.

## Results

In the framework of our proposed model, the family originally contains a planar ejection with strong correlations in the *a-e-i* distribution of family members (see Fig. 1). Whether by KCTF or another means, destabilization of this proto-binary is most likely to happen soon after Haumea reaches the hot classical belt. During this time, Neptune is still completing its final stages of migration, which has not been accounted for in previous models. Our results show that as long as Neptune gets a modest (~0.05–0.1) eccentricity kick after the formation of the Haumea family, the planar distribution is mixed sufficiently to be similar to the presently-observed Haumea family. Modern migration models developed independently to explain other features of the Kuiper belt have such eccentricity kicks and these same models mix the Haumea family enough to produce the uncorrelated *a-e-i* distribution that is observed while maintaining its compact size. The minimal influence of the tail-end of Neptune migration on the compact nature of the Haumea family allows far more flexibility in explaining the family forming collision. The timeframe for this proposed Haumea family formation is potentially quite long based on current models of the formation of the trans-

Neptunian belt, which we adopt without modification in our proposed model.

**Numerical integrations**. Using the n-body integrator REBOUND[28], we have performed a suite of integrations recreating some models of Neptune migration found in the literature. Crucially, many of these models[26,29,30] show Neptune having an instability where the orbital elements can abruptly change in short amounts of time. These abrupt changes, often called jumps, can be modeled via instantaneous changes in the orbital elements of Neptune. In our integrations, we test whether jumps in semi-major axis and eccentricity can enhance mixing.

In each of these integrations, a prototypical planar family was integrated along with the outer planets while Neptune migrated outwards. The initial state of the family is shown in Fig. 1. Upon qualitative analysis of our integrations, it is clear that mixing of the family does indeed occur without excessive erosion. Figure 1 shows the initial state of the family, immediately after creation and when compared to Fig. 2, it is clear that the diffusion of family members is extensive.

**Mixing mechanisms**. While it is clear that our integrations demonstrate the feasibility of mixing due to Neptune migration, it is not immediately clear why the family members are mixed so efficiently. To find the specific mixing mechanisms, we performed some additional integrations to determine the exact mixing mechanisms (see "Methods" subsection "Numerical integrations").

The most intuitive mechanism for mixing before the jump in orbital elements is the transport of the semi-major axes of family members within mean-motion resonances (MMRs). Objects that are captured in MMRs during Neptune's migration are pushed to higher semi-major axes. At the same time, these captured family members typically diffuse to higher eccentricity and lower inclination, as has been seen in many previous analyses. When our final integrations are compared with the exploratory integrations, it is clear that resonant capture, transport, and subsequent removal from MMRs are not responsible for the bulk of the mixing.

The majority of mixing in our simulations occurs after Neptune's jump, which causes a period of enhanced mixing. While not immediately clear whether the jump in semi-major axis or eccentricity is responsible, our exploratory integrations definitively showed that the jump in eccentricity is the dominant factor. Dynamically, the increased eccentricity of Neptune has strong effects on both the strength and size of resonances. Higher-order resonances of the form $p + q$:$p$, several of which are located near the Haumea family, are composed of $q$ subresonances with strengths proportional to $e_N{}^j e^k$ where $j + k = q$. Prior to Neptune's jump, its eccentricity is small and only the $k = q$ subresonance is active for family members near MMRs. Increasing Neptune's eccentricity activates the other subresonances, enhancing chaotic diffusion, leading to the period of mixing after a jump in Neptune's eccentricity.

Combined with the numerical integrations discussed here, our exploratory integrations showed that smooth migration at low eccentricity is probably not sufficient to mix a planar family. Despite this, we expect that many types of Neptune migration models could be capable of mixing a planar family, including ones where Neptune doesn't experience any jumps. Some of our preliminary integrations had Neptune cross a MMR with another planet, temporarily increasing its eccentricity, and subsequently efficiently mixing a planar family. Alternatively, a hard dynamical instability[30] may have a tail end where Neptune's eccentricity is sufficiently elevated to mix a family that forms while Neptune's eccentricity is still non-zero. Despite the wide variety of models,

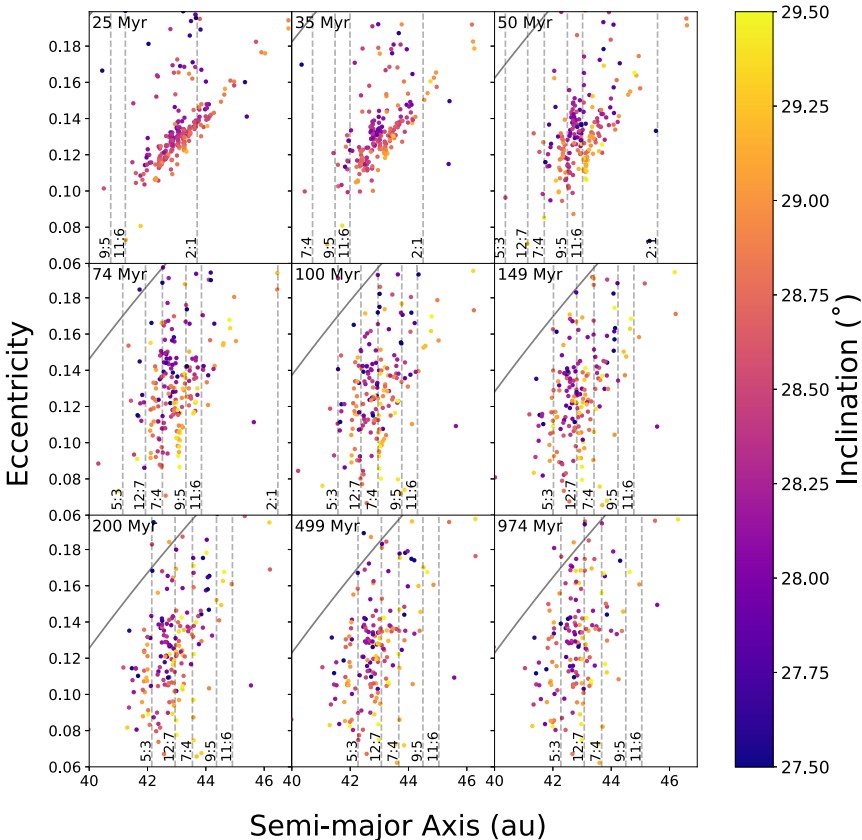

**Fig. 2 Mixing of family members during migration.** The averaged orbital elements throughout our integration. The orbital elements are found using a 50 Myr centered moving average of the instantaneous orbital elements of each object; this corresponds roughly to proper elements for non-resonant objects, though technically proper elements are not well-defined for migrating planets. Each panel is labeled with the time in the top left, with Neptune's jump occurring at 35 Myr. Note that the color bar deviates slightly from Fig. 1. In each frame, dashed, gray lines show the instantaneous locations of some of Neptune's mean-motion resonances (MMRs). The diagonal, solid black line is an estimate of where scattering with Neptune becomes strong enough to remove objects (7/6 $a_N$, corresponding to 35 au with $a_N = 30$ au). For a comparison to the family without migration, see Fig. 1. In the first panel, objects which have been captured into the 2:1 MMR are clearly migrating to higher eccentricities and lower inclinations (darker colors). In the second panel, at the time of the jump, the objects which were previously inside the 2:1 MMR have now been dropped out of resonance. Additional resonances passing through the family create chaotic diffusion in eccentricity over the next several panels, removing some of the *a-e-i* correlation and changing the tilted elliptical shapee5. Throughout this process, objects near resonances sometimes have their eccentricity excited, moving them into the unstable region where scattering becomes dominant. By the end of the integration, the original *a-e-i* correlation present in the planar family has been substantially obscured. Source data for this figure are provided as a Source data file.

many have periods of increased $e_N$ which is thought to be an essential component of successfully reproducing features in the Kuiper belt[31]. Thus we believe that the results presented here could be reproduced with different migration scenarios, though we hope that future observations of Haumea family members will be able to place specific constraints on outer solar system formation models.

**Quantitative assessment of the mixing.** In addition to qualitative assessment, the mixing efficiency was also evaluated using the state-of-the-art Bayesian fitting routine outlined in Proudfoot and Ragozzine[3]. This fitting routine takes an observed family and determines posterior distributions for the properties (location, extent, angular dispersion, etc.) of a model family designed to match a variety of formation hypotheses, without any dynamical evolution. While these fits produce posterior distributions for over a dozen parameters, we focus on the angular dispersion parameter $\kappa$ as a measurement of the planar-ness of the family. When $\kappa > 250$, the family has an angular dispersion <5°, which we call planar. We find that only ~3.5% of the resulting posterior distribution was consistent with a planar ejection, while the vast majority of the posterior was consistent with an isotropic (non-

planar) ejection. The rejection of a planar ejection at a ~2-$\sigma$ level and the distinct shape of the posterior distribution are both extremely similar to results found in previous work based on the present-day observed family[3]. In Fig. 3, we compare these distributions. In Fig. 3, all the posterior distributions show a large peak at $\kappa \sim 10$. This has been previously explained as a result of overfitting, but may indicate that the model is trying to reproduce the boxy shape of the family produced by Neptune migration.

**Erosion and expansion of the family.** We measured the extent of the erosion and expansion of the family after being subjected to Neptune's migration. We found that 35–40% of family members are removed by Neptune's effects. Given the current estimates for the mass of the family (~3% of Haumea's mass[7]), we estimate that the mass of the Haumea family was initially ~5% of Haumea's mass. This is in closer agreement with smoothed particle hydrodynamic models which estimated a family mass of ~7% of Haumea's mass, given a graze-and-merge formation[9]. We do note, however, that the real-world strengths of each MMR passage are likely underestimated, and the number provided here is likely a lower bound on the mass removed.

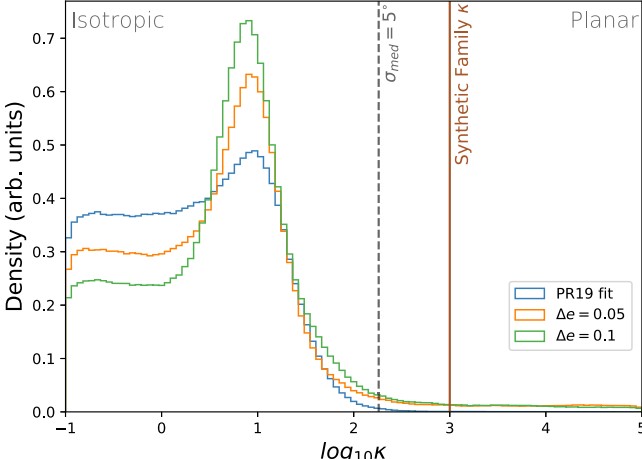

**Fig. 3 Comparing our integrations to the true family.** A comparison between the posterior probability distribution of $\kappa$—the planar concentration parameter—of our two integrations (in orange and green) and the $\kappa$ distribution from PR19 found for the true family (in blue). Marked with a gray dashed line is the value of $\kappa$ above which a synthetic family could be explained by a graze-and-merge collision ($\sigma \lesssim 5°$). We also mark in brown the value of $\kappa$ which the planar family was created with. Only ~3.5% of the posterior distribution of our integrations is consistent with a graze-and-merge formation, similar to the 1% found for the actually observed Haumea family. All three posteriors display a large peak near $\kappa = 10$, which is attributable to overfitting. This demonstrates that Neptune migration can mix a graze-and-merge family into an $a$-$e$-$i$ distribution similar to the observations, though we emphasize that it is not equivalent to fitting the proposed model to the observational data. Source data for this figure are provided as a Source data file.

Despite the removal of much of the family, the family does not expand a great deal. We find that the median $\Delta\mathbf{v}$ of family members is approximately doubled, but most family members retain a $\Delta\mathbf{v} < 150$ m s$^{-1}$, as is observed in the current family (see "Methods" subsection "$\Delta\mathbf{v}$ comparisons"). This is partly due to the Haumea family's proximity to orbits that are long-term unstable due to Neptune interactions.

## Discussion

A variety of observational constraints suggest a graze-and-merge collision origin for the Haumea family: very small ejection velocities, Haumea's unusually rapid rotation, and the extreme water ice spectra of family members. Previous works' primary objection to the graze-and-merge collision was its ~2.5-$\sigma$ inconsistency with the observed $a$-$e$-$i$ distribution of family members[3] (and all other hypotheses were also rejected). By adding the expected mixing due to Neptune migration, we have shown that the expected shape of the family is consistent with the observations, turning this weakness into a strength. As a result, the graze-and-merge formation hypothesis, augmented with a collision of a proto-binary, satisfactorily matches all the observational constraints without invoking improbable events, unlike previously proposed hypotheses. In addition, our integrations clearly show that the Haumea family, after formation near its current location, can survive some forms of Neptune migration. This opens up the possibility that family formation mechanisms besides our proposed mechanism could conceivably match the known constraints on the Haumea family. In the future, the loosened constraints on the timing of the family formation should be incorporated into proposed mechanisms.

Family models without Neptune migration already suffered from the small number of known Haumea family members. Attempting to match post-migration models to the observed family would have been computationally challenging and underconstrained. However, as the Vera C. Rubin Observatory's Legacy Survey of Space and Time (LSST) is expected to discover and characterize ~80 new Haumea family members within the first ~2 years of the survey, future analyses could potentially identify properties of the original ejection of family members. For example, future work could identify a subset of family members relatively unaffected by Neptune migration that retain the original planar ejection characteristics.

At present, our results do not provide strong evidence for one migration scheme over another. In testing, we found that several different schemes worked well to mix the family, including a jumping Neptune with moderate eccentricity jumps, a four-planet model with a period of excited eccentricities due to resonances between the outer planets, and increased eccentricity with Neptune at its current location. Our integrations are not all appropriate for the actual solar system but are varied enough to show that mixing is not an unusual outcome. With more known family members and detailed modeling, the shape and size of the Haumea family may provide valuable constraints on models of Neptune migration.

Future work should also explore the details of various components of this hypothesis such as the frequency of proto-binaries large enough to explain Haumea; geophysical evolution of the interiors of proto-Haumea binary components and Haumea itself; formation of Haumea's moons from ejected debris; expected family ejection directions generated by binary collision (possibly caused by KCTF); relative chronology of Haumea's formation within the phases of Neptune migration; new hydrodynamics simulations of relevant graze-and-merge collisions; and comparisons to other mantle-stripping collisions. For example, a differentiated proto-Haumea should have had a crust of other volatiles, which crust appears to be missing from present family members. Can hydrodynamical models explain what happened to this crust? Was it volatilized and thus absent never formed into solid family members? Are these crust pieces much darker and thus simply harder to find in present surveys?

Despite our simple integrations, which neglected to model some of the complexities involved in the phase of giant planet migration, our integrations show, as a proof-of-concept, that compact families can persist throughout the final migration of Neptune. Further works should explore the survivability/mixing efficiency when more realistic conditions are added. Some of these could include simultaneous migration of the other planets (Jupiter, Saturn, and Uranus), effects of an inclination jump during the dynamical instability, differing migration timescales, differing eccentricity damping timescales, and others.

Another important effect that was not taken into account was the collisional evolution of the family after formation, both from family member–family member and family member–interloper collisions. However, given the shallow size distribution of the family[3,7], there is no evidence for significant collisional grinding after the creation and mixing of the family. This may be due to the lack of major collisions (which are quite improbable) or the low observability of sub-families. Even if evidence for collisional grinding was present, we believe that it would likely only enhance the effective mixing. Each collision would create more (sub-)family members with a spread in $a$-$e$-$i$ space, enhancing the mixing of particles when taken as a whole, although it may lead to an even more mass-depleted family. One interesting avenue of research would be to look for sub-families (or pairs) among the members of the Haumea family to find evidence of any putative collisional evolution.

While the majority of the confirmed family lies within $\sim 150\ \mathrm{m\ s^{-1}}$, there are two spectrally confirmed family members which lie further from the family, 1999 $OY_3$ and 2003 $SQ_{317}$ in addition to Haumea itself. The previous work[5] showed that these objects could diffuse in nearby MMRs, thereby reducing their $\mathbf{\Delta v}$ to be well within the family. However, more accurate orbit determinations, alongside new dynamical integrations, show that only Haumea is presently affected by MMRs: 1999 $OY_3$ is 0.6% wide of the 7:4 and 2003 $SQ_{317}$ is 1.2% wide of the 5:3. While unexpected in previous hypotheses, these two objects can be easily explained in the framework of our integrations. In both of our integrations, at the time of the dynamical instability, many family members begin diffusing chaotically in eccentricity. This chaotic diffusion lasts for a short time while Neptune's eccentricity is still excited. Once the period of diffusion ends, the eccentricity distribution of the family has broadened significantly, leaving a fuzzy edge to the family at high eccentricities, especially near present-day MMRs. The bulk of the synthetic family members remains between about $e = 0.1$–0.145, while the edges of the e-distribution extend to $e = 0.08$ and 0.18. This naturally explains the distribution of confirmed family members, which has a similar morphology. Identifying larger numbers of family members should reveal this fuzzy edge which may help to confirm/constrain the formation mechanism we have proposed.

In the broader context of family finding in the Kuiper belt, our results may hold a clue as to why other large Kuiper belt families have yet to be identified. In our integrations, we find that 35–40% of family members are removed from the family; most of these removed family members are eventually ejected from the solar system, helping to keep the compositional signature of Haumea family members confined to one region in a-e-i space. This confinement is due to the Haumea family's (otherwise unrelated) proximity to the perihelion stability limit ($q \sim 35$ astronomical units [au]), below which Kuiper belt objects (KBOs) quickly become unstable. If the Haumea family was formed at lower eccentricity, the effects of Neptune's migration (and jump) would cause the family to expand greatly in eccentricity, with far fewer family members being ejected from the solar system. While the circumstances of the Haumea family's formation obviously contribute to its detectability (large, bright family members with extremely unique surfaces), its location near the perihelion stability limit also likely plays a role in its early identification. Other (currently hypothetical) KBO families formed at this time may have been significantly diluted by Neptune's migration, rendering them undetectable.

In summary, we have identified a formation mechanism that can match all known aspects of the Haumea family. We propose that Haumea and the family were formed in the aftermath of a binary collision. This family, due to the conditions of the graze-and-merge collision, was ejected at low velocity ($\mathbf{\Delta v} \sim 150\ \mathrm{m\ s^{-1}}$) in a planar ejection pattern. This ejection pattern, which is not found in the observed family members, was subsequently erased by Neptune's outward migration. Our integrations outlined here show that the mixing of a family from Neptune migration is a common and expected outcome in the final stages of planetary migration, despite previously held beliefs that Neptune migration would destroy the family. Even though we found a significant mixing effect, the family is not excessively eroded or expanded. We expect that these results and conclusions will significantly shape any future study of the Haumea family, and may even help to place constraints on Neptune's final stages of migration.

## Methods

**Numerical integrations**. In our numerical integrations, the motions of the outer planets are tracked, along with 250 test particles representing a simulated graze-and-merge family. The Neptune migration model we use follows Nesvorny[26]. This model has Neptune migrating outwards and having a discontinuous jump at ~28.0 au. It can be easily parameterized to create any size jumps in Neptune's orbital parameters $a_N$ and $e_N$. This allows for easy comparison to previous works, where testing in a very similar manner was done.

This jump, or more accurately a mild instability, is thought to be required to create the so-called kernel in the cold classical population[26]. The jump in these models is the consequence of a planet–planet scattering event after Neptune has ejected another ice giant planet out of the solar system, which has been shown to reliably create a final solar system architecture similar to ours today[29]. The existence of such a jump has been supported across a variety of works[27,31–33].

To set up these integrations, Jupiter, Saturn, and Uranus are placed on orbits with semi-major axes and inclinations equal to their current semi-major axes and inclinations. We place Neptune interior to its current orbit ($a_{N,0} = 26.0$ au) with zero inclination. All the outer planets are started with zero eccentricity. In addition to the outer planets, we place a prototypical graze-and-merge type family into the integrations as test particles.

Our numerical integrations rely on REBOUND[28], using the WHFAST symplectic integration scheme[34]. In it, we insert additional forces[35] to migrate Neptune's semi-major axis and damp its eccentricity on a single e-folding timescale, $\tau = 50$ Myr. The timescale used here is similar to timescales shown to match the properties of the outer solar system[26,27]. Neptune was migrated outwards until $a_N \sim 28.0$ au. We then instantaneously change Neptune's orbit such that $a_N = 28.5$ au and $e_N = 0.05$ or 0.1. This brackets the range of $\Delta e$ that was found to be suitable for producing the Kuiper Belt kernel. After this change, the integration was allowed to continue until the total duration was 1 Gyr. In each case, multiple runs were considered, with each run adjusting the migration amplitude so that Neptune's final semi-major axis was within 1% of its current semi-major axis. This was specifically done to best reproduce the locations of Neptune's MMRs with respect to the family. The most promising integrations were extended in a pure n-body model by 4.5 GYr to show how the family would appear today. These long-term integrations showed very little additional evolution except for some resonant diffusion, consistent with previous analyses[36,37].

One shortcoming of our integrations was the non-realistic outer planet eccentricities. Jupiter, Saturn, and Uranus were placed on circular orbits to reduce the chances of massive dynamical instabilities among the outer planets. When tested against integrations with realistic eccentricities, the integrations were almost identical. As Neptune is subject to eccentricity damping throughout the integrations, the coupling between the eccentricities of Jupiter/Saturn/Uranus and Neptune was broken. This allows for the survival of the family through Neptune's migration, even during the strong resonance sweeping the family is subjected to. Furthermore, only Neptune's eccentricity is important for Kuiper Belt dynamics. We remind the reader that these integrations are a proof of concept to show that the Haumea family could have survived Neptune's migration.

In addition to these integrations with a jumping Neptune, we also completed several exploratory integrations to determine the dominant mechanism for family mixing observed in the other integrations. These were not designed to match existing proposed Neptune migration schemes, unlike our nominal model. We had three classes of integrations to test this.

First, we completed many integrations where we have Neptune migration but no jump in eccentricity. Initial conditions of Neptune were the same as above, with the inclusion of a jump of 0.5 au when Neptune reached ~28.0 au. This determines whether a jump in the semi-major axis only is responsible for the mixing observed in our jumping Neptune model.

Secondly, we performed several integrations with a smoothly migrating Neptune with no jumps in semi-major axis or eccentricity. Allowing us to determine if smooth migration was key to the mixing.

Lastly, we completed integrations with eccentricity jumps without Neptune migration. In these integrations, Neptune has its current semi-major axis but was started with $e = 0.1, 0.05, 0.025$. This (somewhat) separates the effect of heightened eccentricity in the immediate aftermath of Neptune's jump from both the migration of Neptune and the jump in Neptune's semi-major axis. Eccentricity damping was implemented to match Neptune's current eccentricity of near zero.

Using these additional integrations, we found that the majority of mixing was caused by increased eccentricity immediately after Neptune's jump. While the semi-major axis jump did enhance the mixing when compared to smooth migration, it was not as clear as the eccentricity-driven mixing.

**Synthetic families**. In each integration, a simulated graze-and-merge family was inserted. To facilitate comparison between integrations, the same family was used in all the integrations. This family was chosen as a representative and likely example of a graze-and-merge collision that would be relatively difficult to mix. Using a family generation method, as outlined in Proudfoot and Ragozzine[3] (hereafter PR19), we created a family consisting of 250 simulated family members ejected with a planar concentration parameter corresponding to a vertical dispersion from a plane of ~2° and with collision center orbital elements (43.1 au, 0.125, 28.2°). The method described in PR19 takes the collision center orbital elements, along with a number-size-velocity distribution for ejected family members and creates a simulated collisional family. The other parameters used to specify the number–size–velocity distribution of simulated family members are $\alpha = 0.2$, $\beta = 0.1, S = 0.8, k = 1.5$, and $\lambda = 1.7$; see PR19 for full details.

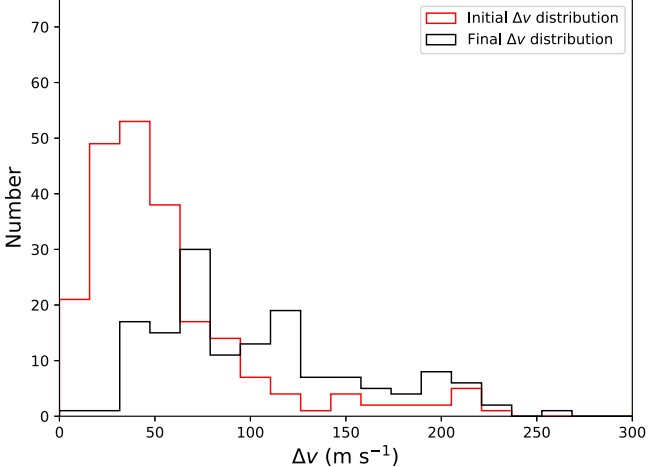

**Fig. 4 Dispersion of family members during Neptune migration.**
Displayed are histograms of the final Δ**v** distribution (in black) compared to the initial Δ**v** distribution (in red) for our integration. As the Δ**v** distribution was found based on the 50 Myr averaged orbital elements, the initial distribution already deviates slightly from the original distribution. During the period of chaotic diffusion in the first few hundred Myr of our integrations, family members tend to increase their Δ**v**, resulting in the distribution shown where the median Δ**v** is about twice the initial median Δ**v**. Source data for this figure are provided as a Source data file.

Although all of the integrations we show in this work contain the planar prototype family, many of the exploratory integrations, as well as preliminary runs of our a, e jump integrations were completed with other realizations of a graze-and-merge family. Throughout our testing process, while using other graze-and-merge families, some families were easier to mix than others, but we conclude that a wide variety of graze-and-merge families are susceptible to mixing through Neptune migration. The chosen family represents a fairly typical graze-and-merge family, with the planar ejection direction chosen to create an initially strong *a-e-i* correlation.

**PR19-style testing**. To determine whether Neptune migration erased the *a-e-i* correlations of a graze-and-merge family, we fit each family using the methods of PR19. That method uses a Bayesian parameter inference framework to infer orbital elements of the collision center, the number-size-velocity distribution of the ejection, and the shape of the ejection field. This is done by creating synthetic families, characterizing them with multivariate normal distributions, and comparing them to a random set of 22 family members chosen from our integrations. The synthetic families are parameterized by 13 parameters (3 parameters describing the planar shape of the family, 5 parameters for the orbital elements of the collision center, and 5 parameters describing the number-size-velocity distribution of family members following Lykawka et al.[38]. The key parameter that is important to our analysis is the angular dispersion parameter, $\kappa$, which characterizes the isotropy/planar-ness of the family. Practically everything about these fits was identical to the method in PR19. As our model is composed entirely of known family members, we do not include the interloper fraction used in PR19. We use the same priors, a $10^5$ step burn-in, and $10^5$ step sampling which showed excellent convergence. For a more in-depth treatment of these methods, see PR19.

**Δv comparisons**. We compare the Δ**v** distributions of the family at the earliest point in the integration with the Δ**v** distribution at the end of the integration. To do this, we do not a priori choose the collision orbital elements, as there is significant uncertainty to the true collision orbital elements of the Haumea family. Instead, we choose the collision orbital elements by minimizing the sum of the Δ**v** of each family member, similar to the previous works[5]. This comparison is shown in Fig. 4, for the integration. The comparison is extremely similar to our other preliminary integrations. In both integrations, the Δ**v** distribution is somewhat broadened throughout the integration, roughly doubling the median Δ**v** of the family, but keeping most within Δ**v** < 150 m s$^{-1}$ as is observed.

## Data availability

All data used and generated in this work have been permanently stored and backed up on the authors' local drives and backups. This includes Simulation Archive files containing all details of our simulations and chain files produced in our PR19-style analysis. While these data are not stored publicly due to their large sizes, it is available to anyone, without condition, upon request from the corresponding author. Source data are provided with this paper.

## Code availability

REBOUND (and its integrator WHFAST) is publicly available code made available at https://github.com/hannorein/rebound. All other codes, including plotting codes, integration codes, and PR19 testing codes, were custom developed for this work. It is available, without conditions, upon request from the corresponding author.

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

## Acknowledgements

We gratefully acknowledge Nate Benfell for his work on backwards integration of the Haumea family. We also thank Steven Maggard for his orbital integrations to find proper elements for the family. We thank Steve Desch for useful discussions. We gratefully acknowledge funding support of the NASA Solar Systems Workings Program under grant 80NSSC19K0028 for B.P.

## Author contributions

B.P. performed all integrations, analysis, and coding required for this work. D.R. advised in all activities and helped with writing, editing, and interpreting results.

## Competing interests

The authors declare no competing interests.
