## [Peer Review File · Nature Communications]

REVIEWER COMMENTS

Reviewer #1 (Remarks to the Author):

I read the revised manuscript and the response to reviewers, and, while the authors have taken care to address many of my points raised in the original review, I do not think that the main issue has been resolved in a satisfactory manner. This issue is the effect of the 2:1 resonance passage on the Haumea family.

The manuscript dedicates lots of space to the comparisons of the structures of synthetic families in the orbital element space to that of the real family. I understand that the authors think that this structure is a crucial bit of evidence that can reveal the origin of the family. However, I think that the authors are missing the big picture, which is that this paper aims to revise the established opinion in the field that the Haumea family had to post-date Neptune's migration because it would have been otherwise obliterated.

Right now the manuscript is not at all clear about the rate of survival of family members in case of realistic sweeping by the 2:1 resonance. For along time the community thought that such sweeping would completely erase the family. If it does not, there has to be some general explanation of where Haumea comes from originally (as it is a dynamically hot object), and how much of the family (and the background) are lost in this resonance sweeping. To reiterate: I am less concerned with there being enough dynamical excitation to scramble the family (the authors have established that there likely was), but I am more worried about the family not surviving in sufficient numbers a more realistic migration of Neptune. I understand that the authors want to stay focused on the Haumea family itself, but unfortunately I do not think that is fully possible given the amount of constraints we have on the Kuiper Belt. In any case, the current amount of added discussion on the dynamical excitation in a non-circular model of the planetary system is not sufficient to justify the authors' conclusions.

Beyond this one issue, I think that the manuscript is fit for publication.

Reviewer #2 (Remarks to the Author):

The submitted manuscript tackles one of the most puzzling problems in planetary science: the format of the system of the dwarf planet Haumea, often called the Haumea 'family'. This terminology is not strictly correct under a dynamical point of view, but is acceptable in a context in which it is clear that a group of bodies have similar orbital elements due to a plausible collisional origin.

The manuscript seems to have included most of the requests of former reviews on previous submission to another journal of the Nature series. In particular, it seems to have successfully overcome one of its weakest points: the non-flatness of the inclination distribution of the family. The authors explain in a satisfactory way how further analysis circumvented that issue.

The authors clearly state that they do not try to reproduce every feature of the Haumea system, instead they concentrate on demonstrating its origin as a former binary collision as a 'proof of concept'. Under this point of view, this reviewer considers that they successfully reached their goal and they put together a very well-motivated demonstration of their thesis.

Therefore, this reviewer recommends the manuscript for publication as is.

Referee Response

We again thank the two anonymous reviewers for the efforts in reviewing our manuscript. The additional feedback has helped us to refocus our manuscript and hopefully makes our paper even more convincing. We have responded, as last time, point-by-point and will provide a manuscript with the changes highlighted.

Referee #1 Response

I read the revised manuscript and the response to reviewers, and, while the authors have taken care to address many of my points raised in the original review, I do not think that the main issue has been resolved in a satisfactory manner. This issue is the effect of the 2:1 resonance passage on the Haumea family.

The manuscript dedicates lots of space to the comparisons of the structures of synthetic families in the orbital element space to that of the real family. I understand that the authors think that this structure is a crucial bit of evidence that can reveal the origin of the family. However, I think that the authors are missing the big picture, which is that this paper aims to revise the established opinion in the field that the Haumea family had to post-date Neptune's migration because it would have been otherwise obliterated.

Response: We thank the reviewer for pointing out that our paper can challenge the current belief that the Haumea family must pre-date Neptune migration. While we have mentioned this throughout the manuscript, we have not focused on this with as much strength as we ought to. In response to this, we have added several statements which make this clearer. These can be seen on page 1 (abstract), page 8, and page 10.

Right now the manuscript is not at all clear about the rate of survival of family members in case of realistic sweeping by the 2:1 resonance. For a long time the community thought that such sweeping would completely erase the family. If it does not, there has to be some general explanation of where Haumea comes from originally (as it is a dynamically hot object), and how much of the family (and the background) are lost in this resonance sweeping. To reiterate: I am less concerned with there being enough dynamical excitation to scramble the family (the authors have established that there likely was), but I am more worried about the family not surviving in sufficient numbers a more realistic migration of Neptune. I understand that the authors want to stay focused on the Haumea family itself, but unfortunately I do not think that is fully possible given the amount of constraints we have on the Kuiper Belt. In any case, the current amount of added discussion on the dynamical excitation in a non-circular model of the planetary system is not sufficient to justify the authors' conclusions.

Response: We agree with the reviewer that survival of the family under the influence of a 2:1 resonance passage is not trivial. As mentioned previously, our preliminary integrations showed that survival of the family was relatively trivial, however, we did not produce any evidence of this in our response or in our manuscript, which was a mistake. To remedy this, we have completed a new set

of integrations in which the outer planets (except Neptune) are started on their current orbits, and all other integration parameters remain relatively unchanged. Qualitatively, these integrations show little to no increased disruption from the 2:1 resonance when compared to the integrations shown in our manuscript. Along with this response and the updated manuscript, we have attached an animation showing the evolution of the family with realistic outer planet eccentricities.

While the survival of the family through a 2:1 resonance passage may, at first, seem unlikely, we believe that there are several reasons for the survival. Firstly, our simulations migrate Neptune outwards on a near-circular orbit, as the included eccentricity damping efficiently breaks the coupling between the outer planets' eccentricity. We did this, as it seems likely that the final portion of the outward migration of Neptune was circular due to the scattering of planetesimals (Nesvorny 2015, and many others). With Neptune on a near circular orbit, resonance passages are far weaker, allowing a good survival rate for the family. Another factor is Neptune's jump, which causes the 2:1 resonance to ~instantaneously jump ~0.8 au. This quickly drops out all objects within the resonance and works to reduce the damage when compared to an integration with a full, uninterrupted 2:1 passage. Despite these protections, the 2:1 passage is still a time of instability within the family. Below, is the first frame of our integration, which shows several family members being removed by the 2:1 passage (with even more removed objects out of the frame). While these family members end up far from the family, they all end with perihelia that result in strong dynamical interactions with Neptune, which eventually ejects them from the solar system. To make our conclusions in the manuscript stronger, we have included some discussion of this on page 11. We have also added some comments on page 10 discussing the broader implications of this when applied to other (currently hypothetical) KBO families. We have also moved the paragraph discussing how the 2:1 passage could be more destructive than shown in our integrations to the methods section.

Although, we believe that our response above addresses the concerns about the 2:1 resonance passage, the initial emplacement of Haumea on its dynamically excited orbit has not been addressed. As detailed in the excellent review by Morbidelli & Nesvorný 2019 (chapter 2 in *The Trans-Neptunian Solar System*), the hot population is thought to have been emplaced when Neptune migrated through the primordial disk of material exterior to Neptune with resonances and resonant drop-outs a central part of this process. If we assume that the proto-Haumea was emplaced prior to the passage of the 2:1 resonance, as illustrated in our integrations, a resonance exterior to the 2:1 must have taken part in its emplacement, with some candidates being the 3:1, 5:2, or 9:4. While further investigation of the emplacement process could produce meaningful constraints on the process, we feel that emplacement prior to the 2:1 passage is not so unlikely as to place our model in jeopardy. Even so, we have added a more in-depth discussion of the emplacement process to our manuscript, visible on page 3.

Beyond this one issue, I think that the manuscript is fit for publication.

Response: We thank the reviewer for seeing the merit of our work and for helping us to improve the quality and strength of this paper.

In addition to the changes outlined here, small procedural changes have been made to grammar, syntax, etc. All of these changes are visible on the document with the changes shown in green.

Referee #2 Response

The submitted manuscript tackles one of the most puzzling problems in planetary science: the format of the system of the dwarf planet Haumea, often called the Haumea 'family'. This terminology is not strictly correct under a dynamical point of view, but is acceptable in a context in which it is clear that a group of bodies have similar orbital elements due to a plausible collisional origin.

The manuscript seems to have included most of the requests of former reviews on previous submission to another journal of the Nature series. In particular, it seems to have successfully overcome one of its weakest points: the non-flatness of the inclination distribution of the family. The authors explain in a satisfactory way how further analysis circumvented that issue.

The authors clearly state that they do not try to reproduce every feature of the Haumea system, instead they concentrate on demonstrating its origin as a former binary collision as a 'proof of concept'. Under this point of view, this reviewer considers that they successfully reached their goal and they put together a very well motivated demonstration of their thesis.

Therefore, this reviewer recommends the manuscript for publication as is.

Response: We thank the reviewer for their review of our updated manuscript and our response to their previous comments. Our manuscript has been made significantly stronger with their helpful comments.

REVIEWERS' COMMENTS

Reviewer #1 (Remarks to the Author):

The authors have addressed my previous comments in a satisfactory manner, and I think that the manuscript is now ready for publication.